# Independence Threat or Interdependence Threat? The Focusing Effect on Social or Physical Threat Modulates Brain Activity

**DOI:** 10.3390/brainsci14040368

**Published:** 2024-04-09

**Authors:** Guan Wang, Lian Ma, Lili Wang, Weiguo Pang

**Affiliations:** 1The School of Psychology and Cognitive Science, East China Normal University, Shanghai 200062, China; 2School of Education Science, Huaiyin Normal University, Huaian 223300, China; 3School of Computer Science and Technology, Huaiyin Normal University, Huaian 223300, China

**Keywords:** emotion threat, action threat, ERP, physical threat, social threat

## Abstract

Objective: The neural basis of threat perception has mostly been examined separately for social or physical threats. However, most of the threats encountered in everyday life are complex. The features of interactions between social and physiological threats under different attentional conditions are unclear. Method: The present study explores this issue using an attention-guided paradigm based on ERP techniques. The screen displays social threats (face threats) and physical threats (action threats), instructing participants to concentrate on only one type of threat, thereby exploring brain activation characteristics. Results: It was found that action threats did not affect the processing of face threats in the face-attention condition, and electrophysiological evidence from the brain suggests a comparable situation to that when processing face threats alone, with higher amplitudes of the N170 and EPN (Early Posterior Negativity) components of anger than neutral emotions. However, when focusing on the action-attention condition, the brain was affected by face threats, as evidenced by a greater N190 elicited by stimuli containing threatening emotions, regardless of whether the action was threatening or not. This trend was also reflected in EPN. Conclusions: The current study reveals important similarities and differences between physical and social threats, suggesting that the brain has a greater processing advantage for social threats.

## 1. Introduction

The ability to detect and process various threats is the most important cognitive function for humans [1,2]. Humans are social creatures, and effective socialization with caregivers as children can enable them to provide better care. Therefore, the ability to understand and respond to social threats is essential for individual survival. As individuals continue to grow, acquiring the ability to recognize new threats, such as knives, guns, and violent actions, is necessary to adapt to a changing environment. There are also studies that refer to both threats as phylogenetic and ontogenetic threats [3].

Focusing on different threats may bring completely different results. For example, individuals with anxiety disorders are more likely to pay attention to emotional faces, whereas boxers instead pay more attention to their opponent’s swinging fists. In everyday life, different types of threats are presented simultaneously and attract our attention. Some studies have focused on how emotional faces and body postures attract individuals’ attention [4,5], but as of now, it is largely unknown whether these different types of threats interact with each other, or even who has a greater processing advantage. Whether there is a threat-specific cognition over the different types of threats or whether threat is organized in a central or in a distributed fashion in the human nervous system are all questions that currently exist.

The threat of complexity corresponds more closely to the actual phenomena people encounter in their social lives. Threatening messages from other people in your life consist primarily of perceptual social threat signals, such as fearful faces, or physical threats, such as angry body movements. They are sometimes separate, but most of the time they occur simultaneously. Depending on their research needs, researchers will categorize these messages not only as emotional messages but also as threat messages when needed [6]. This research orientation clearly has some limitations and ambiguities. Some researchers have proposed that faces or body movements embody an emotion–intent duality [7,8]. This means that these stimuli convey both emotional and threat information. Therefore, if researchers are unable to effectively separate emotions from threats and conflate the two, it can greatly limit researchers’ understanding of threats and emotions.

Neuroanatomical evidence suggests that the human brain has specialized pathways to rapidly process threatening stimuli. Amygdala circuits are especially important for threat detection [9]. It has been found that the human brain processes threat emotions in a top-down manner mediated by the prefrontal cortex and bottom-up processing responsible for the amygdala [10]. Following rapid assessment, the amygdala is involved in re-feedback to the temporal and occipital cortex, which manifests itself in enhanced responses to emotional faces [9]. Several studies have revealed in detail the circuits that enable rodents to respond to unconditioned threat stimuli (US), such as odors that signal predators or potentially dangerous animals [11]. These odors are detected by the plow-nose olfactory system and sent to the medial amygdala (MEA). In addition to olfaction, unconditioned threats delivered through other modalities (e.g., visual auditory stimuli) need to be transmitted from the sensory system to the lateral amygdala [12]. The above studies seem to demonstrate the importance of the amygdala in threat processing.

Although amygdala-centered neural circuits have important advantages for threat processing, according to the sensory account of threat processing [13], the sensory cortex does not require the assistance of the amygdala for the processing of certain threats. The sensory cortex is not thought to be passively involved in threat processing, nor is it fully subservient to the commands of structures such as the amygdala. The theory suggests that when the sensory cortex encodes a threat, the memory encoding of the threat is activated, and the initial threat assessment is triggered during the sensory feed-forward scan, without even requiring the amygdala and associated threat processing networks. This is because plasticity in the sensory cortex develops from aversive experiences (e.g., conditioned threat) that evolve over time into long-term memory traces that become acquired associative representations (AARs) of threat cues. Subsequent encounters with threat cues activate the corresponding AARs, driving rapid and accurate threat assessment in the sensory cortex. It has been found that the olfactory sensory cortex, but not the amygdala, is the region that encodes the overall response to acquired threats (AARs) and plays a key role in threat memory [14]. The above suggests that individuals’ processing of certain types of threat (e.g., conditioned threats based on aversive experiences) stimuli occurs without feedback from the subcortical amygdala.

In real life, it is often the case that social and physical threats are presented simultaneously, and angry faces are often accompanied by aggressive body movements. Few studies have explored which party holds a greater processing advantage in the amygdala and sensory cortex during complex threat cognition, and what the mechanisms of their interaction are. The mechanisms by which the amygdala and sensory cortex act on each other during threat information processing have become a hot topic pursued by many researchers. However, most studies have used functional MRI to conduct relevant research, and conclusive electrophysiological evidence in humans is scarce.

Compared to functional MRI, event-related potentials have very high temporal resolution and can accurately measure millisecond-level potential changes in the brain after an event. This allows researchers to analyze the brain’s rapid response processes to different stimuli or tasks. Electrophysiological (mainly event-related potentials) studies of threatening emotions have outlined the time course of threatening emotion processing, and there is a general consensus on the temporal stages of emotion processing. First, as indexed by the P1 component (~100 ms), visuosensory processing of emotional stimuli occurs in the occipital cortex. Second, as indexed by the N170/N190 component (~170 ms) [15], which represents the rapid structural encoding of emotional faces and body movements. Most of the multiple studies recorded directly by humans demonstrate that the amygdala has a latency of more than 160 ms to threatening emotional stimuli [16]. Subjective judgements can be made not only about stimulus features but also about emotions. This suggests that social threat processing begins at 160 ms. According to the sensory account of threat processing [13], the human sensory cortex can drive intelligent (fast and precise) assessment of acquired threats. There is an early stage of emotion regulation with specific components of EPN. EPN, which is a negative-going wave arising around 150–300 ms after picture onset, is known to reflect automatic selective visual attention toward emotional stimuli [7,8]. Relative to neutral stimuli, threat pictures of different intensities elicit greater amplitude. Unlike the amygdala, which is initially sensitive to the processing of threat stimuli (initial significance), the enhancement of threat stimuli by EPN is not altered by habituation [17], and thus this component may be more sensitive to learned conditioned threats.

In contrast to innate fears, learned fears are experience-dependent and can develop throughout the entire lifespan. In order to explore the characteristics and temporal dynamics of amygdala and visual cortex responses to threat information, we presented participants with two types of pictures at the same time, one of threatening emotional faces (social threat, a typical unconditioned stimulus) [18] and one of fighting actions (physiological threat, a conditioned stimulus based on the aversion conditioning learned) [19]. The current study recorded electroencephalography (EEG) signal characteristics when participants attended to pictures of threatening emotional faces (e.g., anger) or fighting movements (e.g., striking). It hypothesized that N170 and EPN are important electrophysiological indicators of social threat processing, as evidenced by larger wave amplitudes in the angry emotion condition than in the neutral emotion condition. Significant electrophysiological markers of processing physiological threats include N190 and EPN, as demonstrated by higher wave amplitudes in the fighting action group compared to the non-fighting action group. In addition, because of the amygdala’s processing advantage for emotional faces, which can be followed by automated processing of threats and then involvement in re-feedback to the temporal and occipital cortex, the present study hypothesized that social threats affects the processing of physiological threats when faces and actions are present simultaneously under the action judge task. As a result, images containing threatening emotions elicited higher N190s and EPN, regardless of whether the images contained fighting actions.

## 2. Materials and Methods

### 2.1. Participants

We calculated the sample size before the experiment using G-power; we set the effect size f value as 0.25, the α value as 0.05, and the power value as 0.80. The sample size was 24. Twenty-five college student volunteer participants participated in the experiment. Ten males and fifteen females, with normal vision or corrected vision, all right-handed, without any cognitive or neurological impairment, aged 18–27 years (*M* = 23, *SD* = 1.8) were involved, and all participants were unclear about the purpose of the experiment. At the end of the experiment, subjects received a material reward (CNY 80–100) depending on their accuracy. The study was conducted under the guidance of the Declaration of Helsinki. The Huaiyin normal university ethics committee approval number is HNU-TSS-20240313-1.

### 2.2. Experimental Materials

The current study chooses emotional faces as social threats, and the selected materials are from the Nimstim face emotion picture library [20], which has been widely cited by emotion-related research papers and has high reliability and validity. We selected two categories of emotion pictures from this library, one for angry emotions and one for neutral emotions. Ten pictures of each type of emotion were selected, and then the emotional identity represented by each picture was measured. Twenty college students were recruited and asked to judge the emotional identity of each picture, all with 100% emotional identity.

Fighting movements were selected as physiological threats for the current study. First, three types of action pictures were created (fighting threat/gymnastics–non-threat/walking–neutral). Thirty-five body movement pictures were selected for the stimulus materials, 11 for the walking body movements and 12 each for the gymnastics and fighting movements, and about half of the stimulus materials were images of men and women, and the movement pictures were downloaded from various websites. Taekwondo, Boxing, and Kick Boxing were used for the fighting movements; gymnastics movements were used for the non-threatening movements. The physical characteristics of these gymnastics movements were similar to those of the fighting movements in that they involved stretching of the limbs and large body movements. Using walking body movements as the target task could avoid the influence of novel category stimuli. To ensure ecological validity, the stimulus materials were real people with backgrounds removed and faces mosaiced.

To ensure that the body movement pictures in this study were not emotional, a current study compared them with emotional body movements. For this study, 30 emotional body movement pictures (10 each of anger, fear, and happiness) were collected from the BEAST (Bodily Expressive Action Stimulus Test) database [21], processed identically, and mixed with the previous 35 pictures. Twenty psychology graduate students were asked to categorize the emotional body movements based on anger, fear, and pleasure. It was found that the 30 emotional body movement pictures from the BEAST database and another 5 produced movement pictures were excluded because they carried emotions or could easily be misclassified as emotional movements. Finally, 30 stimulus pictures (10 walking, 10 fighting, and 10 gymnastics) were left, and the evaluators’ non-emotional agreement with these pictures was 91.63%, 92.15%, and 90.08%, respectively. Next, a pre-test allowed 25 psychology students to rate the threat and friendliness of these body movement pictures. A 7-point threat rating scale was used to assess the threat dimension; similarly, a 7-point friendliness rating scale was used to rate friendliness. A pseudo-randomized order was used for stimulus presentation. As expected, the ANOVA results showed a significant main effect of threat ratings for different physical movements, *F* (2,47) = 148.81, *p* < 0.001, *η*^2^ = 0.71, and fighting movements (*M* = 5.90, *SE* = 0.23) were perceived to be more threatening than walking (*M* = 2.58, *SE* = 0.23, *p* < 0.001) and gymnastics (*M* = 2.30, *SE* = 0.23, *p* < 0.001). Similarly, there was a significant main effect of friendliness ratings for the different body movements, *F* (2,47) = 96.80, *p* < 0.001, *η*^2^ = 0.68. Gymnastics movements (*M* = 5.88, *SE* = 0.31) were perceived as the more friendly than fighting movements (*M* = 2.52, *SE* = 0.25, *p* < 0.001), whereas the same was true for walking movements (*M* = 5.35, *SE* = 0.30, *p* = 0.07), which did not differ. There was no difference in arousal between the fighting and gymnastics maneuvers. This result suggests that the fighting movements conveyed a threat signal and the gymnastics movements, similar to the walking movements, did not convey a threat signal. Ultimately, we chose the two types of movements, fighting and gymnastics, for inclusion in the experiment.

### 2.3. Procedure

The experiment was conducted in a muffled and semi-darkened room. Participants sat comfortably on an armchair in front of a 17-inch monitor, and in order to ensure that their responses to various stimuli would not be affected by attentional effects, they were approximately 50 cm away from the monitor, and two images were located on the top and bottom of the screen, with emotional faces on the top and body movements on the bottom, and the overall size of the images was 5 × 5 cm, with an viewing angle of 10.0° × 4.9°, and the image sizes were 256 × 512 pixels, and the image sampling rate was 72 pp. The experiment was divided into two parts, the first part was called the action judgement task, and the second part was called the emotion judgement task. In the first action judgement task, a “+” gaze point appeared at the bottom of the screen at the beginning of each trial, which lasted 1500 ms, and the participants were asked to stare at the “+” when it appeared to ensure that their attention was shifted to the corresponding position on the screen according to the requirements. The brain waves triggered by the vertical eye movement of the participants were kept at a stable level. Next, a picture of an action appeared immediately at this position, and an emotional face appeared at the top of the screen, both lasting for 500 ms (see Figure 1). The subject’s task was to decide whether it was a fighting or non-fighting action within 2500 ms, and if it was a fighting action, press 5; if it was a non-fighting action, press 2. They were instructed to ignore the picture of the emotional face appearing at the top of the screen in the meantime and move on to the next trial. In the second emotion judgement task, a “+” gaze point appeared at the top of the screen at the beginning of each trial, and the participant had to stare at this gaze point when the “+” appeared to ensure that the attention was shifted to the corresponding position on the screen as required. Next, a picture of an emotional face appeared immediately at this location, and the task was to determine whether it was an angry or neutral emotion by pressing the number 6 key if it was an angry emotion or pressing the number 3 key if it was a neutral emotion, again ignoring the picture of a body movement that appeared at the bottom of the screen during this process. The task order and response buttons were counterbalanced between participants, and the other half of the participants pressed the opposite buttons. The response screen disappeared after the key was pressed. Each task was preceded by a practice phase consisting of 20 random trials with an accuracy of 70% or more before being allowed to enter the formal experiment. In the formal experiment, each task consisted of 240 trials = 30 cycles × 2 (upper and bottom) × 4 (combinations: angry expression–fighting action, angry expression–gymnastics action, neutral expression–fighting action, neutral expression–gymnastics action). There was a break of about 5 min between tasks, during which participants were asked to read an excerpt from Zhu Ziqing’s essay “Back Shadow”, which describes nostalgia for one’s father. Participants were asked to maintain attention to the “+” gaze point and to avoid blinking or moving their head. The action pictures appeared with the same probability, the different categories of emotion pictures appeared with the same probability, and the order of appearance of all combinations of action and emotion faces was randomized. The final data were analyzed by extracting only the trials with the correct response.

After recording the event-related potentials (ERPs), participants were shown the same set of pictures again and asked to rate each of the two types of threats (social and physical). Intensity was rated out of 10 (1 = “no threat”; 10 = “high threat”). One week prior to the start of the ERP recording experiment, participants completed a series of personality disposition questionnaires: an emotional susceptibility questionnaire and threat sensitivity (THT+) to access sensitivity to emotion and threat perception tendencies. The emotional susceptibility questionnaire consists of 25 questions [22] and is scored on a 4-point Likert scale (never, seldom, often, and always), with a Cronbach’s alpha coefficient for the questionnaire of 0.90 and the trait fear inventory [23] is used to access sensitivity to threat sensitivity (THT+).

### 2.4. Data Recording and Analysis

EEG data were collected using the NeuroScan Synamps 2 EEG recording and analysis system (Neuroscan, Sterling, VA, USA), which was used to present the stimuli through E-Prime 2.0. EEG data from 62 electrode positions on the scalp as well as Horizontal Electrooculogram (HEOG) and Vertical Electrooculogram (VEOG) were recorded using a 64-conductor Ag/AgCl electrode cap, and the electrodes were arranged according to the international 10–20 standard. The left mastoid was used as the reference electrode, the contralateral mastoid as the recording electrode, and the forehead was grounded. Electrodes were placed at the lateral canthus of both eyes to record horizontal electrooculograms (HEOGs), and electrodes were placed above and below the left eye to record vertical electrooculograms (VEOGs). AC sampling was used, with a sampling frequency of 500 Hz and a filtered bandwidth of 0.01–100 Hz. The scalp resistance was less than 5 kΩ, and both mastoids were averaged as a reference for offline analysis; an FIR digital filter was used to perform a phase-shift-free low-pass digital filtering at 30 Hz (24 dB/oct) to exclude EEG events with myoelectricity and drift, and then to refer to the subjects’ eye movements. After excluding obvious EMG events with EMG and drift, the subjects were referred to the eye movements, and the electrooculograms were corrected by independent component analysis (ICA) to exclude artefacts exceeding ±80 μV after baseline correction. The EEG was analyzed over a period of 1000 ms, of which 200 ms before stimulus presentation was used as the baseline. Data pre-processing was performed in MatlabTM 9.0 (2016).

Based on the previous arguments, in the emotion judgement task, we calculated the mean wave amplitude P1 for occipital lobe O1 and O2 electrodes between 100 and 120 ms and performed a two-factor repeated-measures ANOVA: 4 (angry face/fighting action, angry face/gymnastics action, neutral face angry/action, and neutral face/gymnastics action) × 2 electrode positions (O1, O2) in the experimental design. Mean wave amplitude N170 was calculated for the occipitotemporal lobe PO7 and PO8 electrodes between 145 and 175 ms, and wave amplitudes were analyzed by two-way repeated measures ANOVA for each of the following: 4 (angry face/fighting movements, angry face/gymnastics movements, neutral face/angry movements, and neutral face/gymnastics movements) × 2 electrode positions (PO7, PO8) for the 4 × 2 experimental design. Mean wave amplitude EPN was calculated for the occipital–temporal lobe PO7 and PO8 electrodes between 240 and 330 ms, and wave amplitudes were analyzed by two-way repeated measures ANOVA separately: 4 (angry face/fighting action, angry face/gymnastics action, neutral face/fighting action and neutral face/gymnastics action) × 2 electrode positions (PO7, PO8) for a 4 × 2 experimental design. The Greenhouse–Geisser method was also used to correct the *p* values. Post hoc comparisons were performed using the Newman–Keuls test.

In the action judgement task, the current study analyzed P1, N190, and EPN, and the present study calculated the mean wave amplitude N190 of the occipital–temporal lobe PO7 and PO8 electrodes between 155 and 185 ms for a within-subjects two-factor repeated measures ANOVA: 4 (angry face/fighting action, angry face/gymnastics action, neutral face/fighting action and neutral face/gymnastics action) × 2 electrode positions (PO7, PO8) for the experimental design; the other components and the emotion judgement task were calculated identically. Correlation and regression analysis was performed to assess whether the subjective evaluation of threat intensity is different from the change in neural activity under different tasks. To calculate reaction time and accuracy, the present study used a within-subjects two-factor repeated measures ANOVA: 4 (stimulus type: angry face/fighting action, angry face/gymnastics action, neutral face/fighting action, and neutral face/gymnastics action) × 2 (task type: emotional task/action task) for the experimental design. In all cases, these components were statistically evaluated using SPSS (version 22.0) and Greenhouse–Geisser correction was used when the hypothesis of sphericity was not met. For significant main effects or interactions, Bonferroni corrected *p*-values were reported for post hoc comparisons.

## 3. Results

### 3.1. Electrophysiological Analysis

Under the emotional judgement task, the total mean ERPs and topographic maps under the co-stimulation of emotion and action are displayed in Figure 2. The descriptive statistics of the components in each of the two tasks are shown in Table 1.

At P1, the two-factor repeated-measures ANOVA results showed that *F* (3,72) = 14.68, *p* < 0.001, partial *η*^2^ = 0.38. Multiple comparisons showed that the amplitude of the wave was significantly higher for NFE than for AFE [*t* (72) = 4.64, *p* = 0.001, cohen’d = 0.55] and the AGE [*t* (72) = 5.03, *p* < 0.001, cohen’d = 0.57]. Similarly, the wave amplitude was significantly higher for NGE than for angry face/fighting action [*t* (72) = 4.52, *p* = 0.002, cohen’d = 0.53] and AGE [*t* (72) = 4.17, *p* = 0.002, cohen’d = 0.55]. The difference between NFE and NGE on P1 was not significant [*t* (72) = 0.15, *p* = 1, cohen’d = 0.02]. The difference between AFE and AGE was also not significant [*t* (72) = 0.09, *p* = 1, cohen’d = 0.05]. This result suggests that the visual cortex of the brain at the P1 stage can recognize simple stimuli and does not readily process complex stimuli. The main effect of P1 in the left and right brain regions was not significant, *F* (1,24) = 0.98, *p* = 0.33, partial *η*^2^ = 0.04.

The N170 components differed significantly across conditions. Two-factor repeated-measures ANOVA results showed that *F* (3,72) = 10.52, *p* < 0.001, partial *η*^2^ = 0.31. Follow-up multiple comparisons showed that the wave amplitude was significantly higher for AFE [*t* (72) = 4.87, *p* < 0.001, cohen’d = 0.10] than NFE [*t* (72) = 4.50, *p* = 0.001, cohen’d = 0.14] and NGE [*t* (72) = 3.55, *p* = 0.01, cohen’d = 0.10]. The difference between the amplitude of the wave for the AGE and the neutral face/fighting action [*t* (72) = 1.15, *p* > 0.05, cohen’d = 0.03] was not significant, and the difference between the amplitude of the wave for the NGE [*t* (72) = 0.29, *p* > 0.05, cohen’d = 0.55] was not significant. The difference between NFE and NGE on N170 was also not significant [*t* (72) = 1.38, *p* = 1, cohen’d = 0.04]. Overall, the N170 wave amplitude was greatest for AFE, and the difference in wave amplitude was not significant in the other conditions. The main effect of N170 in the left and right brain regions was not significant, *F* (1,24) = 1.09, *p* = 0.31, partial *η*^2^ = 0.04.

EPN components differed significantly across conditions. Two-factor repeated-measures ANOVA results show that *F* (3,72) = 23.64, *p* < 0.001, partial *η*^2^ = 0.47. Subsequent multiple comparisons showed that the wave amplitude was significantly higher for AFE than for NFE [*t* (72) = 13.32, *p* < 0.001, d = 0.46] and NGE [*t* (72) = 9.64, *p* < 0.001, cohen’d = 0.36]. Similarly, the wave amplitude was significantly higher for AGE than for NFE [*t* (72) = 4.74, *p* < 0.001, cohen’d = 0.47] and NGE [*t* (72) =3.40, *p* = 0.014, cohen’d = 0.36]. The difference between AFE and AGE on EPN was not significant [*t* (72) = 0.04, *p* = 1, cohen’d = 0.01]. This result under the emotion judgement task shows that anger triggered a greater wave amplitude in EPN than in the neutral emotion group, regardless of the action.

Under the action judgement task, the total mean ERPs and topographic maps are displayed in Figure 2. Descriptive statistics of the components concerned in each of the two tasks are shown in Table 1. At P1, two-factor repeated-measures ANOVA results showed that *F* (3,72) = 91.32, *p* < 0.001, partial *η*^2^ = 0.79. Subsequent multiple comparisons showed that the amplitude of the wave was significantly higher for NFV [*t* (72) = 8.44, *p* < 0.001, cohen’d = 0.43] than for AFV [*t* (72) = 28.65, *p* < 0.001, cohen’d = 0.93]. Similarly, the wave amplitude was significantly higher for NGV than for AFV [*t* (72) = 4.91, *p* < 0.001, cohen’d = 0.42] and AGV [*t* (72) = 12.95, *p* < 0.001, cohen’d = 0.92]. The difference between NFV and NGV was not significant [*t* (72) = 0.08, *p* > 0.05, cohen’d = 0.01]. AFV was significantly higher than AGV [(*t* (72) = 9.17, *p* < 0.001, cohen’d = 0.44)]. This result combined with the results of the emotion judgement task, in terms of the results of the present study, suggest that P1 is responsible for the initial processing of simple conditioned stimuli or intrinsically consistent complex conditioned stimuli. Furthermore, the main effect of P1 was significant in left and right brain regions, *F* (1,24) =7.99, *p* < 0.01, partial *η*^2^ = 0.25. The left brain wave amplitude (*M* = 3.88, *SE* = 0.37) was smaller than that of the right brain (*M* = 4.42, *SE* = 0.35).

The N190 components differed significantly across conditions. The two-factor repeated-measures ANOVA results showed that *F* (3,72) = 8.43, *p* < 0.001, partial *η*^2^ = 0.26. The amplitude of the waves was greatest for AFV and AGV. Follow-up multiple comparisons showed that the difference in wave amplitude between AFV and AGV was not significant [*t* (72) = 0.82, *p* = 1, cohen’d = 0.07]. However, AFV was significantly higher than NFV [*t* (72) =10.33, *p* < 0.001, cohen’d = 0.29] and NGV [*t* (72) = 4.59, *p* = 0.001, cohen’d = 0.21]. Similarly, AGV was significantly higher than NFV [*t* (72) = 2.95, *p* = 0.041, cohen’d = 0.23], but not significantly different from NGV [*t* (72) = 1.42, *p* = 1, cohen’d = 0.15]. This result is similar to the results of the emotion judgement task, suggesting that N190 may be affected by emotional faces.

The EPN components differed significantly across conditions. The two-factor repeated-measures ANOVA results showed that *F* (3,72) = 22.36, *p* < 0.001, and partial *η*^2^ = 0.48. Follow-up multiple comparisons showed that the wave amplitude was significantly greater for AGV [*t* (72) = 6.29, *p* < 0.001, cohen’d = 0.70] than for AFV [*t* (72) = 4.52, *p* = 0.001, cohen’d = 0.68], and significantly greater than for NGV [*t* (72) = 8.43, *p* < 0.001, cohen’d = 0.97]. The wave amplitude of NFV was significantly greater than that of NGV [*t* (72) = 3.02, *p* = 0.036, cohen’d = 0.43], whereas the difference in wave amplitude between NGV and AFV was not significant [*t* (72) = 0.62, *p* = 1, cohen’d = 0.10]. This suggests that EPN can handle the processing of threatening actions, but there is a phenomenon of interference of emotions on threatening actions. Another result to note is that the wave amplitude of the AFV was significantly smaller than that of the AGV [*t* (72) = 6.29, *p* < 0.001, cohen’d = 0.81]. This may reflect an inhibitory effect between the different threat types.

### 3.2. Questionnaire Results and Ratings

No extreme values were found in the results of the emotional susceptibility questionnaire (*M* = 34.44, *SD* = 2.99) and THT+ (*M* = 53.96, *SD* = 5.81) tests (see Table 2). Individuals did not differ significantly in their perception of threat intensity for anger (*M* = 6.32, *SD* = 0.95) and fighting action (*M* = 6.20, *SD* = 0.71). The paired-sample *t*-test results showed that *t* (24) = 0.45, *p* = 0.66, Cohen’s d = 0.14. All participants finished the stimuli in the task during the ERP recording sessions. Through the paired-sample *t*-test, the results showed no significant difference in task judgement accuracy between the two tasks (emotion judgement task *M* = 94%, SD = 5%; action judgement task M = 92%, SD = 7%), *t* (24) = 0.37, *p* > 0.05, Cohen’s d = 0.15. The one-way ANOVA results at the time of response show a significant main effect, *F* (3,96) = 7.31, *p* < 0.001). The reaction time under the emotion judgement task was smaller than the action judgement task. The reaction time for the anger expression (*M* = 487.04, *SD* = 55.73) was significantly shorter than for the fighting action [(*M* = 543.79, *SD* = 59.69, *t* (72) = 3.56, *p* = 0.001, cohen’d = 1.02] and the gymnastics action [*M* = 541.07, *SD* = 56.03, *t* (72)= 3.39, *p* = 0.001, cohen’d = 0.97], and the response times for neutral expressions (*M* = 492.43, *SD* = 54.18) were also significantly shorter than for the fighting actions [*t* (72) = 3.22, *p* = 0.002, cohen’d = 0.95] and gymnastics actions [*t* (72) = 3.05, *p* = 0.003, cohen’d = 0.90]. There was no significant difference in the reaction time of different emotion types.

### 3.3. Correlation and Regression between Subjective Rating and Event-Related Potential Amplitude

In the AFE task type (*r* = −0.68, *p* < 0.001, R^2^ = 0.13) and the AGE task type (*r* = −0.75, *p* < 0.001, R^2^ = 0.15), there was a significant negative correlation between threatening emotion rating and the N170 amplitudes at PO7 and PO8 (Figure 3). Further, the Fisher test showed that the correlation between AFE’s N170 amplitude and subjective ratings of threat intensity was not significantly greater than that of the AGE (z = 0.45, *p* = 0.66).

The ENP amplitude difference of the PO7 and PO8 electrode in the NFV task type (*r* = −0.93, *p* = 0.001, R^2^ = 0.63) was negatively correlated with threatening movement subjective ratings, whereas no correlation was found in the AFV task type (*r* = −0.01, *p* = 0.93, R^2^ = 0.01) (Figure 3). Further, the Fisher test between the types found that the correlation was significantly stronger for the NFV than the AFV (z = 5.82, *p* < 0.001). The larger ERP amplitude of threat response in the later stage was related to the stronger subjective threatening rating elicited by the fighting action, mainly from the NFV task type.

## 4. Discussion

In summary, the present study recorded and analyzed the EEG activity of participants under different task demands when emotional faces and body actions were presented simultaneously. The present study focused on three components, EPN, N170/N190, and P1. This study showed that the wave amplitude of N170 and EPN in the occipital–temporal lobe can respond to an individual’s processing of threatening emotions under an emotion judgement task. In contrast, the results changed under the action judgement task, with pictures containing threatening emotional faces but not the fighting action pictures, triggering greater N190 activity, as well as changes in EPN results. First, the amplitude of the fighting action was significantly more negative (NFV > NGV) than that of the gymnastics action in the neutral emotion, suggesting that EPN may be a processing indicator of conditioned threat. Second, the wave amplitude of ADV was significantly more negative than NDV and NFV, suggesting that an emotional face may have an effect on action judgement. Again, looking at EPN, the results of this task found that the test with incongruent emotion and action valence was significantly more negative than that with congruent emotion and action valence. This suggests that EPN is a further processing mechanism of the threat stimulus.

Earlier studies have identified that N170 may play a role in the early perceptual processing of emotional faces in individuals. It is used to measure early perceptual encoding and face categorization [24,25]. N190 represents the early encoding of body structures by the human brain [26]. N170 and N190 are both ERP components associated with visual perception. Both components are thought to reflect configural processing mechanisms involved in the perception of faces and bodies, respectively. The current study found similar characteristics of wave amplitude differences across the four stimulus types, both under the face emotion judgement task (N170) and the threat action judgement task (N190), mainly in the form of significantly larger wave amplitudes in the angry emotion group than in the neutral emotion group, regardless of the action type, and with no differences prior to the AFV and AGV, and no significant differences between the NFV and NGV. This result suggests that N170/N190 encode emotions. This is consistent with previous research, which asked highly anxious and lesser anxious individuals to recognize angry expressions of different intensities in an emotion recognition task, and the results support the idea that N170 is a categorical processing mechanism of emotional faces without regard to intensity [27]. While previous studies have argued that N190 is an early structural encoding mechanism of threatening action information, in the action judgement task of the current study, the characteristics of the individual’s early perception of threatening actions (N190) are clearly at odds with a previous research study [28], which used a passive perception paradigm to find that fighting actions triggered N190 significantly greater than neutral and dancing actions. The current study suggests that emotional faces already interfere with threatening action pictures in the early stages of an individual’s processing of stimulus pictures. It is again validated that facial emotion processing is a fast and highly automated process. People rely on the rapid recognition of facial emotion—a social threat—in adaptive life engineering [29]. In addition, body movement research has found that emotional information from both the face and the body interacts, with body actions and postures tending to accentuate and intensify the emotion expressed by the face [26], further suggesting a processing advantage for threat emotions.

Another possibility for this result to occur is because individuals have a processing advantage for unconditioned threats compared to conditioned threats. One study recorded skin conductance responses to conditioned and unconditioned threat stimuli in 351 participants, including healthy individuals and individuals with anxiety disorders, and found that pathological anxiety was primarily due to enhanced responses to potential, rather than immediate, threats [30]. Unconditioned and conditioned threats also differ in early response pathways [31], and it has been found that because evolutionary threat processing relies on lower-order, phylogenetically conserved neural fear circuits, when evolutionary threats (e.g., snakes, spiders) and modern threats (e.g., guns, knives) have equal valence and arousal, evolutionary threats trigger enhanced amygdala activation [32]. This set of findings is consistent with the preparedness theory [3], which postulates that associative memories between stimuli that threaten survival and frightful experiences in evolutionary history are challenging to forget. It also explains why unconditioned threats have a greater impact on processing than conditioned threats.

In addition, N170/N190 processing of emotional stimuli showed some features of automated processing, as evidenced by the fact that stimuli from the angry emotion group triggered greater N170 amplitudes, regardless of whether the task demanded attention to the emotional face or not. This phenomenon can be attributed to the dual processing theory of the amygdala’s treatment of threat. The dual-pathway model of emotional processing in the amygdala suggests that there is a fast subcortical pathway for coarse processing and a slower cortical pathway for fine processing, and that the existence of the subcortical pathway helps organisms to detect threatening stimuli automatically and quickly in situations where sensory information and cognitive resources are limited. One study examined N170′s enhancement of fearful expressions in healthy controls as well as epileptic patients after unilateral amygdala removal and found that N170 did not change much in individuals whose right amygdala was removed. This result suggests that the amygdala produces an early brain response to fearful faces. This early response is particularly dependent on the right amygdala and occurs at around 170 ms [33]. There have also been studies that have eliminated physical features of the face and controlled for facial images by separating top-down modulation from bottom-up driving, and the results found that N170 was not significant across conditions, also supporting the idea that N170 is an automated indicator of bottom-up face information [34].

Most intriguing were the different features presented by EPN under the two tasks. Consistent with previous research, under the emotion judgement task, EPN represents the processing of the threat level of the object of attention [8,35]. The present study not only replicated but also extended previous work and found that under an action judgement task, individuals’ processing of fighting actions was influenced by the emotion of anger. This suggests an advantage in the processing of social threats compared to physical threats. One study compared the neural response processes of snakes and guns, controlling for the stimulation and potency of the two types of stimuli, and found that snakes as an innate threat triggered a stronger EPN [36]. Another result of interest is that the EPN amplitude was significantly smaller in AFV and NGV than in AGV and NFV, and that the amplitude of the consistent face and action valence wave was significantly smaller than that of the inconsistent valence group. Previous studies have suggested that EPN is a selective attention indicator for perceiving threa, and the higher the level of threat, the greater the amplitude. It has also been found that perceiving different emotional expressions leads to reduced activity in the amygdala and cortex, and consistent emotional expressions enhance corresponding brain activity [37]. All these findings appear at first glance to conflict with the current study. In fact, angry faces and fighting actions, although both negative in valence in the present study, are in fact two completely different types of stimuli, one for potential threat and one for direct threat, one representing anxiety and one fear, with different neural circuits, respectively [38], which may lead to reduced activity in the amygdala and cortex. Meanwhile, the picture stimuli in the AGV and NFV groups in the current study contained gymnastics actions and neutral faces without threat or emotion. Since there was no other type of threat, there was no interaction of different neural circuits, which resulted in significantly larger wave amplitudes in the AGV and NFV groups than in the AFV and NGV groups at this stage of EPN.

This study is dedicated to exploring the interaction of social and physical threats. In fact, these two threats may represent innate threats and conditioned threats based on aversion formation, respectively. Like previous studies, which found that after conditioning (following conditioning), innate threats could be stimulated in both CS+ and CS- conditions, while conditioned threats were absent. This suggests that innate threats and conditioned threats are two parallel processes that activate autonomous responses separately [39]. The present study not only replicated previous research but also further found that innate threat processing is more dominant by directing attention to conditioned threats, while the opposite does not hold true. It has been advocated that individuals’ processing of threat emotion is mainly based on the amygdala, whereas processing of conditioned threats may come from the sensory cortex., e.g., fear conditioning promotes sparser representations of conditioned threats in the primary visual cortex [40]. This is consistent with the sensory account of threat processing. The theory suggests that aversive experiences enable the human sensory cortex to perform intelligent threat assessments, evoking threat processing on distributed neural networks to drive rich, flexible threat responses in humans. Although the two types of threat are parallel processors, there are also many connections between the amygdala and the sensory cortex. Šimić et al. advocate that the central circuitry of the amygdala detects threats in the environment and projects instructive signals to the sensory cortex [9]. Some studies have found that damage or inactivation of the amygdala impedes plasticity in the auditory system [41]. Other studies in humans have found that emotion recognition is not only present in the amygdala but also involves extensive reentrant connections from the amygdala to primary visual and ventral stream areas [24]. A recent study used differential odor-threat conditioning in rats to test the role of basolateral amygdala (BLA) input to the piriform cortex in the acquisition and expression of learned olfactory threat responses and found enhanced connectivity of basolateral amygdala with the primary olfactory cortex [42], with a series of results suggesting that amygdala and the sensory cortical together shape the response to conditioned threats. This result is also consistent with the affective realism hypothesis, which emphasizes that emotions can critically shape what we expect and actually see, hear, and smell in the present moment, e.g., threat-related emotional experiences can contribute to perceiving certain objects as more stressful or threatening.

It has been suggested that the P1 component represents early attention allocation [43]. In the present study, due to the lack of salient information, pictures composed of neutral emotions were more difficult to process compared to pictures composed of angry emotion stimuli, and therefore more attention may be allocated to these pictures to ensure that the subsequent task can be successfully completed, as evidenced by a larger mean P1 amplitude. It has been found that a lack of or inconspicuous face information may result in more ambiguous and difficult to judge body expressions [44]. In addition, the influence of emotion on P1 is reflected by the difference between perceptual analysis and low-level visual feature processing, rather than a difference in actual emotional processing [45], and some perceptual features related to emotion in this expression may lead to differences in visual components. It has also been found that inverted face stimuli elicited larger P1 amplitudes compared to orthogonal face stimuli, also suggesting that the P1 component is an early global response to stimuli [46]. Consequently, the neutral emotion group had a larger wave amplitude than the angry emotion group due to the lack of informational salience in the neutral emotion picture group and the tendency of P1 to process holistic information.

The present study, although the first to explore the processing advantages of different threats, contributes to further understanding of the nature of threats. However; there are some limitations in this study, such as the fact that this study explored the interaction mechanism between social and physiological threats. To consider ecological validity, both emotional faces and fighting actions were used. However, these action pictures have only been considered as conditioned threats based on aversive experiences in others’ research [47]. Future research should allow neutral body actions to be matched with US according to Pavlov’s classical conditioned paradigm and go on to further compare the mechanisms of interactions between innate and conditioned threats while ensuring ecological validity. Secondly, the intrinsic cognitive neural mechanisms of social and physiological threats in this study need to be further demonstrated by appropriate techniques. For example, EEG-fMRI combined with skin conductance responses can be used to examine the neural circuits and interaction mechanisms between the two types of threats. Finally, the relationship between social and physical threats may represent the relationship between anxiety and fear. Future in-depth exploration of the interactions between the two could not only confirm the “two systems” view of fear and anxiety [1] but also improve the assessment and treatment of aberrant behaviors in patients with anxiety disorders and PTSD.

## 5. Conclusions

In summary, this paper explored which threat has a greater processing advantage and how threats interact with each other when they are present at the same time through two tasks. The results found that social threats has a processing advantage over physiological threats and may influence the processing of physiological threats when both threat stimuli are present at the same time under attentional conditions. In contrast, physiological threat processing was not found to influence processing of social threat. This sheds light on future systematic explorations of the interaction mechanisms between different types of threats.

## Figures and Tables

**Figure 1 brainsci-14-00368-f001:**
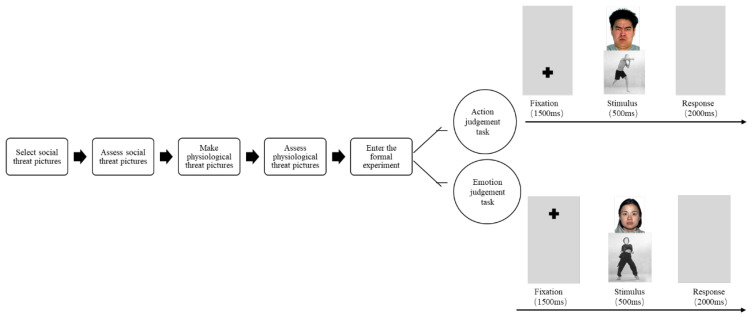
The flow chart of procedure of this study and procedure used in this task shows the sequence of events within a trial.

**Figure 2 brainsci-14-00368-f002:**
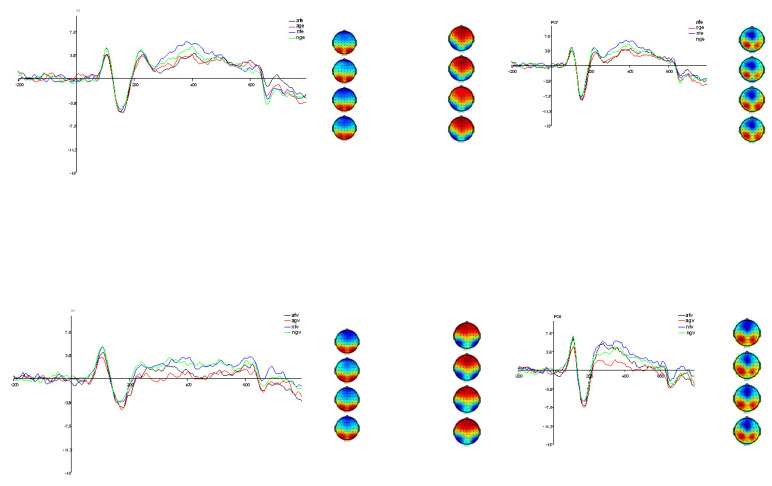
Average ERPs of four conditions and voltage topographies illustrate the scalp distribution for the ERP components. The top left corner shows the average amplitude and voltage topographies of P1 on electrode O1 under four conditions in the emotion judgment task. Thtop right corner presents the average amplitude and voltage topographies of N170 and EPN at the PO7 location under four conditions in the emotion judgment task (topographies: N170 on the left, EPN on the right). The bottom left corner displays the average amplitude and voltage topographies of P1 on electrode O1 under four conditions in the action judgment task. The bottom right corner shows the voltage topographies of N190 and EPN at the PO8 location in the action judgment task (topographies: N190 on the left, EPN on the right). On the topographic map, red represents positive voltage and blue represents negative voltage.

**Figure 3 brainsci-14-00368-f003:**
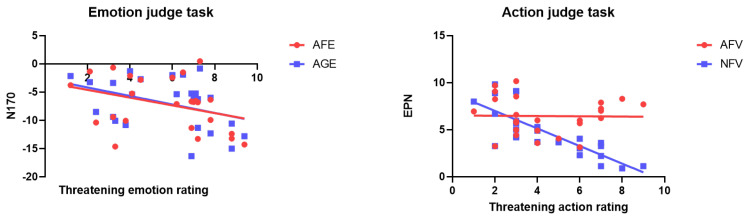
Correlation between the ERP response of face action and subjective threat intensity under emotion judgement task and action judgement task. On the left side, under the emotion judge task, it is indicated that subjective threat assessment is significantly correlated with N170 under AGE and AFE conditions. On the right side, under the action judgement task condition, subjective threat assessment is correlated with EPN under NFV condition, but not related to EPN under AFV condition.

**Table 1 brainsci-14-00368-t001:** Descriptive statistics of each EEG component under two tasks (unit: μv).

Emotional Judgement Task					Action Judgement Task				
Condition	AFE	AGE	NFE	NGE	AFV	AGV	NFV	NGV	
P1	*M* = 3.04, *SD* = 1.65	*M* = 3.26, *SD* = 1.58	*M* = 3.95, *SD* = 1.75	*M* = 3.95, *SD* = 1.32	*M* = 3.67, *SD* = 2.11	*M* = 2.88, *SD* = 1.85	*M* = 4.48, *SD* = 2.17	*M* = 4.50, *SD* = 1.51	P1
N170	*M* = −8.23, *SD* = 5.73	*M* = −7.76, *SD* = 6.47	*M* = −7.36, *SD* = 6.07	*M* = −7.27, *SD* = 5.99	*M* = −6.99, *SD* = 5.06	*M* = −6.87, *SD* = 4.01	*M* = −5.62, *SD* = 4.75	*M* = −5.97, *SD* = 4.76	N190
EPN	*M* = 1.33, *SD* = 3.56	*M* = −1.37, *SD* = 3.54	*M* = 3.00, *SD* = 3.22	*M* = −1.37, *SD* = 3.54	*M* = 4.48, *SD* = 3.15	*M* = 1.45, *SD* = 2.73	*M* = 3.52, *SD* = 2.79	*M* = 5.18, *SD* = 3.17	EPN

(Note: AGE is the abbreviation for angry emotion and gymnastics action group under the emotion judgement task; AFE is the abbreviation for angry emotion and fighting action group under the emotion judgement task; NGE is the abbreviation for neutral emotion and gymnastics action group under the emotional judgement task; NFE is the abbreviation for neutral emotion and fighting action group under the emotion judgement task; AGV is the abbreviation for angry emotion and gymnastics action group under the action judgement task; AFV is the abbreviation for the angry and fighting action group under the action judgement task; NGV is the abbreviation for the neutral emotion and gymnastics action group under the action judgement task; NFV is the abbreviation for the neutral emotion and fighting action under the action judgement task.).

**Table 2 brainsci-14-00368-t002:** Descriptive statistics of questionnaire and ratings.

ESQ	THT+	Threat Perception		Emotion Judgement Task		Action Judgement Task	
		Anger Face	Fight Action	Anger Condition	Neutral Condition	Fighting Condition	Gymnastics Condition
*M* = 34.44, *SD* = 2.99	*M* = 53.96, *SD* = 5.81	*M* = 6.32, *SD* = 0.95	*M* = 6.20, *SD* = 0.71	*M* = 487.04, *SD* = 55.73	*M* = 492.43, *SD* = 54.18	*M* = 543.79, *SD* = 59.69	*M* = 541.07, *SD* = 56.03

## Data Availability

The datasets generated during and/or analyzed during the current study are available from the corresponding author on reasonable request.

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
