# Peer review of "Independence Threat or Interdependence Threat? The Focusing Effect on Social or Physical Threat Modulates Brain Activity"

_brainsci, 2024, doi:10.3390/brainsci14040368_

Round 1

Reviewer 1 Report

Comments and Suggestions for Authors

Review Results

Manuscript entitled: Independence threat or interdependence threat? The focusing effect to social or physical threat modulates brain activity.

I am really appreciative of this study and enjoy reading this manuscript. However, there are some concerns where more details need to be declared, as follows:

Line 33-34: Please delete the following statement:

“We must always control our emotions and not be troubled by them”——Bruce Lee 33 Siu-long”

Line 139: “…, and all subjects were unclear about the purpose of the experiment.

Please kindly explain this statement. What does it mean for “unclear”?   

Line 140: “…end of the experiment, subjects received a material reward (80-100 RMB) depending on their accuracy.” Please kindly explain this statement.

What does it mean for “… depending on their accuracy”?

Line 140-141: The study was conducted under the guidance of the Declaration of Helsinki.

Both EC approval number and date/month/year of approval must be indicated.

2.2 Experimental materials

The flow chart of procedure must be demonstrated so that the readers are able to understand the whole processes of the experiment.

Line 203: “EEG data were collected using the NeuroScan Synamps 2”

The name of company and city must be indicated for EEG and Amplifier devices.

Both EEG and ERP abbreviation must be used only after the full names were indicated.

What was the statistical analysis program used in the data analysis?

How did the mean (+/- SD) of ERP component calculate? Is it at the peak of each ERP component?

References with more than 10 years are not accepted. The authors must revise their references.

Reviewer 2 Report

Comments and Suggestions for Authors

The study examines the brain's response to social and physical threats under different attentional conditions using event-related potentials (ERP). Participants were exposed to facial (social) and action (physical) threats, focusing on one threat type at a time. Findings reveal that the brain processes social threats more dominantly than physical threats, indicating a prioritization of social threat processing. The research contributes to understanding how humans perceive and process complex threats in their environment.

This is a nice paper and I only have some comment.

1) Threatening stimuli are surely of great relevance to humans. In the introduction I would suggest adding a short paragraph to say that threatening stimuli, both represented by faces (Dalmaso et al., 2020; Fox et al., 2002) and bodies (Azarian et al., 2016), can also shape visual attention strongly, as compared to neutral or positive stimuli. This is an important information, which is now missing in the text.

2) Please clarify how you determined sample size (power analysis?)

3) Data sharing: Please provide a link to a database (e.g., OSF) in which future readers can have access to the data.

References

Azarian, B., Esser, E. G., & Peterson, M. S. (2016). Watch out! Directional threat-related postures cue attention and the eyes. Cognition and Emotion, 30(3), 561–569. https://doi.org/10.1080/02699931.2015.1013089

Dalmaso, M., Castelli, L., & Galfano, G. (2020). Social modulators of gaze-mediated orienting of attention: A review. Psychonomic Bulletin & Review, 27(5), 833–855. https://doi.org/10.3758/s13423-020-01730-x

Fox, E., Russo, R., & Dutton, K. (2002). Attentional Bias for Threat: Evidence for Delayed Disengagement from Emotional Faces. Cognition & Emotion, 16(3), 355–379. https://doi.org/10.1080/02699930143000527

Reviewer 3 Report

Comments and Suggestions for Authors

The manuscript analyzes the registration of ERPs with the presentation of drawings modeling physical and social threats. For more than half a century, work with the presentation of various sensory material on a person has remained relevant against the backdrop of solving specific problems. This study is part of a series of similar works. However, the present manuscript requires serious revision. 

1. The manuscript was not prepared in accordance with the preparation format for this journal. 

2. References to literary sources are also not formatted according to the journal’s rules. See rules for authors. 

3. In the Results Chapter there is an application of the Emotional Susceptibility Questionnaire. However, there is no information about this test in the abstract or in the Methods Chapter. 

4. The Results chapter raises the most questions for the specialist. The chapter is oversaturated with textual material listing numerical values. And it contains only 2 figures with its own results. It is customary to accompany such work with ERPs registration with both original data and digital data - figures and tables. Figures and tables should be added to this manuscript. Optimal for components under study and/or test situations. 

5. For the Discussion, as for the Introduction, I strongly recommend considering the data of various authors on waves throughout the epoch of analysis. The authors use an evaluation time of 1000 ms. For example, https://doi.org/10.3390/s22041323. It also makes sense to add an analysis of information on EP registration under the conditions of the oddball paradigm. 

6. The conclusion should be redone based on the results of processing the manuscript.

Round 2

Reviewer 1 Report

Comments and Suggestions for Authors

Review Results: 2nd Round

Manuscript entitled: Independence threat or interdependence threat? The focusing effect to social or physical threat modulates brain activity.

Thanks to the authors for clarifying those of my concerns. However, there are some serious issues to be considered where more details need to be declared, as follows: 

If possible, the authors can combine Figure 1 and Figure 2 to show the whole procedure of the study.

The unit of ERP value must be indicated to the appropriated place in all parts of the manuscript. Without the ERP unit, the reader might not understand the value of all demonstrated data.

The real statistical value of p-value must be indicated. Please avoid mentioning only as p>0.05 or p<0.001.

LINE 280 – 351 must be revised. The current version is not a scientific report. Some parts of information can be reported in the table.

3.2 Questionnaire results and ratings

Question: The authors must keep all data in table so that the readers are able to understand their findings from questionnaire and ratings.  

Figure 3: The color of topography must be indicated. For instance, what does it mean for RED/BLUE color representation?

The high resolution of figure 3 must be shown. The pictures are not so clear enough.

Figure 4 legend must be revised. Precisely context must be applied to avoid confusion.

The authors mentioned in the reply to the reviewer that “The experimental task we carried out is not difficult for the participants, and the accuracy can basically reach 90% after simple training.”

Question: How did the authors know that the experimental task was NOT difficult? Who judge the task difficulty?

Question: How could the authors know that the accuracy can basically reach 90%?

The authors mentioned in the reply to the reviewer that “…. The reward of 80-100 yuan depends on the accuracy of the participants after the completion of the experiment. The higher the accuracy, the higher the reward.”

Question: How much the participants get if they did higher accuracy? This is not clear for me. In addition, it might be unfair from each participant to participant concerning award to receive.

Question: The definition of accuracy is still not clear. How the author compared between 80-100 yuan to the statement of “… the accuracy can basically reach 90%.”

The authors mentioned in the reply to the reviewer that “…. The reward of 80-100 yuan depends on the accuracy of the participants after the completion of the experiment.”

Question: What if the author did not complete the experiment? Did they get the reward?

The authors mentioned in the reply to the reviewer that “…. The reward of 80-100 yuan depends on the accuracy of the participants after the completion of the experiment. The higher the accuracy, the higher the reward. The purpose is to ensure that the subjects can complete the current task seriously and intently.”

Question: How did the authors ensure that the purpose of doing this experiment can cause all participants complete the current task seriously and intently?       

Line 541: Translated with DeepL.com (free version)

Question: I do not understand the above sentence.

----------------

Reviewer 2 Report

Comments and Suggestions for Authors

I am satisfied with this new version.

Author Response

Many thanks to the reviewer for recognizing and encouraging our work.

Reviewer 3 Report

Comments and Suggestions for Authors

Fig. 3 is very small, unreadable. It is necessary to enlarge the captions in this Fig. in the editor.

Table 1. Increase the space between component data. M (plus, minus) SD will also look better.

Author Response

Fig. 3 is very small, unreadable. It is necessary to enlarge the captions in this Fig. in the editor.

Thanks to the reviewer's advice, we have reworked the diagram to enhance the image resolution. Since Figure 1 and Figure 2 were merged as other reviewer‘s suggestion, this modified one is now in order Figure 2.

Table 1. Increase the space between component data. M (plus, minus) SD will also look bette

Thanks to the reviewer's advice, we have optimized Table 1.

Finally, I would like to thank the reviewers once again for every one of the suggestions they have given us for our work, which have been very enlightening.